# Psychometric evaluation of a quality of recovery score for the postanesthesia care unit—A preliminary validation study

Ursula Kahl[1], Katrin Brodersen[1], Sarah Kaiser[1], Linda Krause[2], Regine Klinger[1], Lili Plümer[1], Christian Zöllner[1], Marlene Fischer[1,3]*

1 Department of Anesthesiology, University Medical Center Hamburg-Eppendorf, Hamburg, Germany, 2 Institute of Medical Biometry and Epidemiology, University Medical Center Hamburg-Eppendorf, Hamburg, Germany, 3 Department of Intensive Care Medicine, University Medical Center Hamburg-Eppendorf, Hamburg, Germany

* mar.fischer@uke.de

## Abstract

### Introduction

Patients' perception of postoperative recovery is a key aspect of perioperative care. Self-reported quality of recovery (QoR) has evolved as a relevant endpoint in perioperative research. Several psychometric instruments have been introduced to assess self-reported recovery 24 hours after surgery. However, there is no questionnaire suitable for use in the postanesthesia care unit (PACU). We aimed to develop and psychometrically evaluate a QoR questionnaire for the PACU (QoR-PACU).

### Methods

The QoR-PACU was developed in German language based on the 40-item QoR-40 questionnaire. Between March and November 2020, adult patients scheduled for elective urologic surgery completed the QoR-PACU preoperatively and during the PACU stay. We evaluated feasibility, validity, reliability, and responsiveness.

### Results

We included 375 patients. After two piloting phases including 72 and 48 patients, respectively, we administered the final version of the QoR-PACU to 255 patients, with a completion rate of 96.5%. Patients completed the QoR-PACU at a median of 125.0 (83.0; 156.8) min after arrival in the PACU. Construct validity was good with postoperative QoR-PACU sum scores correlating with age (r = 0.23, 95% CI: 0.11 to 0.35, p < 0.001), length of PACU stay (r = -0.15, 95%CI: -0.27 to -0.03, p = 0.02), pain in the PACU (r = -0.48, 95% CI: -0.57 to -0.37, p < 0.001) and piritramide dose administered (r = -0.29, 95% CI: -0.40 to -0.17, p < 0.001). Cronbach's alpha was 0.67 (95% CI: 0.61–0.73) with moderate test-retest reliability (ICC of 0.67, 95% CI: 0.38 to 0.83). Cohen's effect size was 3.08 and the standardized response mean was 1.65 indicating adequate responsiveness.

**Data Availability Statement:** Data cannot be shared publicly because of the German General Data Protection Regulation. Without patients' consent the regulation does not allow for sharing

data with researchers that are not part of the predefined research team. For further information please refer to the data protection officers of the City of Hamburg (Der Hamburgische Beauftragte für Datenschutz und Informationsfreiheit, Ludwig-Erhard-Str 22, 7. OG, 20459 Hamburg) and the University Medical Center Hamburg-Eppendorf (Datenschutzbeauftragter, Universitätsklinikum Hamburg-Eppendorf, Martinistr. 52, 20246 Hamburg, Tel.: 040/7410-56890, E-Mail: m. jaster@uke.de).

**Funding:** Ursula Kahl was funded by the Clinician Scientist Program of the medical faculty of the University of Hamburg, during the conduct of the study. The University of Hamburg was not involved in any of the following: study design, conduct of the research, preparation of this manuscript, analysis and interpretation of data; writing of the report; decision to submit the article for publication.

**Competing interests:** The authors have declared that no competing interests exist.

## Conclusion

The assessment of QoR in the early postoperative period is feasible. We found high acceptability, good validity, adequate responsiveness, and moderate reliability. Future studies should evaluate the psychometric properties of the QoR-PACU in more heterogeneous patient populations including female and gender-diverse patients with varying degress of perioperative risk.

## Introduction

The improvement of postoperative recovery is a common aim of all disciplines involved in perioperative care [1–3]. Postoperative recovery after surgery and anesthesia has traditionally been assessed using objective parameters including but not limited to cardiovascular, pulmonary or infectious complications, pain or length of hospital stay [4–6]. In recent years patients' perception of recovery after surgery has been increasingly recognized as a relevant outcome measure [1, 2, 7]. To allow for comparability across clinical studies, the *Standardized Endpoints in Perioperative Medicine (StEP)* initiative recommends six standardized outcome measures reflecting patient comfort: postoperative pain, nausea, time to gastrointestinal recovery, time to mobilization, sleep disturbance, and the assessment of postoperative quality of recovery (QoR) [8, 9]. In the same line, the introduction of patient-reported outcome assessments is recommended by the *American Society for Enhanced Recovery and Perioperative Quality Initiative* [10]. Various instruments have been developed to evaluate postoperative patient-reported recovery. Myles and colleagues developed the 40-item QoR-40 questionnaire that has been validated, translated, and used extensively [2, 11–14]. In 2013, the same research group developed the 15-item QoR-15 questionnaire which is a shorter version of the more extensive QoR-40 [7, 15–24]. Both instruments have been introduced to assess QoR one day after surgery [2, 7]. The importance of advanced recovery room care and the assessment of patient-centered outcomes early after surgery has recently been highlighted by an Australian feasibility study [25]. Yet, there is no instrument appropriate for application in the postanesthesia care unit (PACU).

The aim of this study was to develop a QoR questionnaire for the PACU (QoR-PACU) and to evaluate its feasibility, validity, reliability, responsiveness, and clinical acceptability in patients after general anesthesia for elective non-cardiac surgery. Therefore, we developed the German QoR-PACU and evaluated its psychometric properties in a cohort of patients scheduled for elective urologic surgery.

## Materials and methods

### Ethical approval and study registration

Ethical approval was obtained from the ethics committee at the Hamburg State Chamber of Physicians on February 11, 2020 (serial number PV7218). Each patient gave written informed consent before the initiation of study-related procedures. The study was registered at clinical-trials.gov (NCT04528537).

### Study design and participants

We performed a prospective observational cohort study at the Department of Anesthesiology of a tertiary care university hospital in Northern Germany. All study participants completed the QoR-PACU on the day before surgery to obtain baseline values. Patients were assessed

postoperatively 120 minutes after arrival in the PACU allowing for a tolerance interval of ± 60 minutes. A subgroup of 19 patients underwent a second postoperative assessment after another 60 ± 30 minutes to evaluate test-rest reliability. Additionally, test-retest reliability was assessed in a different subset of 44 patients on the first postoperative day. Patients read and completed the questionnaire themselves. If necessary, patients were provided with glasses by the study team. All assessments were performed by three examiners (KB, SK, MF).

Patients were included, if they were 18 years or older and received general anesthesia for elective radical prostatectomy. We excluded patients scheduled for same-day surgery, ambulatory surgery or postoperative admission to the intensive care unit and patients without excellent German language skills.

## Development and adaptation of the QoR-PACU

The aim of the study was to develop a questionnaire derived from the QoR-40 to assess the QoR in PACU. Three experienced anesthesiologists (MF, LN, CZ) independently selected 15 items each from the validated German translation of the QoR-40 [26], which they deemed to be of high clinical importance for recovery and self-perceived health status during the early postoperative period [26]. After thorough discussion, a consensus version containing 16 items was developed (version 1). Similar to the QoR-15 questionnaire, an 11-point numerical rating scale was used with a score from 0 ("none of the time") to 10 ("all of the time"). For negative items, the scoring was reversed. A total score was calculated ranging from 0 to 160 points, with a higher score representing better recovery. We performed a pre-test of the QoR-PACU in a convenience sample of 10 patients to assess feasibility. After a successful pretest phase, we started a pilot testing of the QoR-PACU, aiming to optimize acceptability und feasibility of the questionnaire. During the first pilot testing, we administered the 16-item QoR-PACU (version 1) to 72 patients. However, we repeatedly noticed misunderstandings. One major issue was the 11-point response scale from 0 to 10, reflecting the frequency of positive or negative symptoms. A relevant number of patients was confused with simultaneous pain ratings, which are part of clinical practice in the PACU, and assessed intensity rather than frequency. Therefore, we reduced the 11-point scale to a 5-point scale from 0 to 4. For ease of understanding we linked each number with an adverb of frequency: 4 points = always, 3 points = most of the time, 2 points = occasionally, 1 point = rarely, 0 points = never, resulting in a total score from 0 to 64 points. For negative items, the scoring was reversed. During the second pilot testing, we administered the 16-item QoR-PACU (version 2) to another 48 patients. We noticed that feasibility of the questionnaire was hampered by the font size of the questionnaire. Some patients were not able to read the questionnaire themselves, so the study team read the questions aloud to them. In order to improve feasibility and comparability of the assessment, we enlarged the font size and provided patients with reading glasses. To ensure acceptability of the QoR-PACU, we interviewed patients and PACU staff and consequently reduced the QoR-PACU questionnaire to 13 items, resulting in a total score from 0 to 52 points. Four items were dropped for lack of importance as reported by patients and PACU staff: the distinction between severe and moderate pain, shivering, bad dreams, and the feeling of being alone. The item "nausea and vomiting" was separated into two items. This final version of the QoR-PACU was used in our study (version 3, S1 and S2 Tables).

## Data collection

Medical history and demographic characteristics were collected during the preanesthesia visit. We recorded the following clinical data: age, gender, body mass index, Charlson Comorbidity Index (CCI), obstructive sleep apnea syndrome, medication, American Society of

Anesthesiologists (ASA) physical status classification, education, and current profession. To preoperatively assess the risk for obstructive sleep apnea syndrome we used the STOP-Bang score that evaluates snoring, tiredness, observed apnea, high blood pressure, body mass index, age, neck circumference, and male gender with higher scores indicating a higher risk. We retrieved information about the duration of surgery, length of PACU stay, intra- and perioperative medication from anesthesia protocols. The Numeric Rating Scale (NRS) was used to assess pain intensity in the PACU.

## Sample size

There is no consistent recommendation regarding sample size for the development and the evaluation of a questionnaire. A "rule of thumb" suggests at least 10 participants for each scale item [27, 28]. This would result in 160 participants for the 16-item version 1, or 130 participants for the 13-item version 3 of the questionnaire. We did not a priori decide on the exact number of items to be included in the questionnaire. Instead, we planned to conduct a pilot phase with at least 100 patients to optimize the development of the questionnaire. For the final analysis, we aimed to include at least 200 participants. Since we expected a large drop-out rate of about 20%, we opted for the overall inclusion of 375 patients.

## Statistical analysis

Continuous variables are presented as median with $25^{th}$ and $75^{th}$ percentiles, or as mean ± SD. Categorical variables are given as absolute and relative numbers.

Validity was assessed using the postoperative QoR-PACU sum score. To assess construct validity, we compared postoperative QoR-PACU sum scores between categories of clinically relevant variables using a linear model and analysis of variance. Additionally, we analyzed correlations between postoperative QoR-PACU sum scores and clinically relevant continuous variables using Pearson's product-moment correlation coefficients and corresponding 95% confidence intervals.

Reliability was analyzed using results of individual items of the postoperative QoR-PACU and sum scores of postoperative QoR-PACU of those who took the tests twice postoperatively. Reliability was assessed based on internal consistency using Cronbach's alpha, split-half reliability and test-retest reliability using Pearson's correlation coefficient. Pearson correlation was chosen over the intraclass correlation coefficient since it is reasonable to assume that the state of the patient changed within the one hour of time between both assessments [29].

Cronbach's alpha was calculated between items of the postoperative QoR-PACU using the alpha function from the R-package "psych" version 2.1.9 [30]. An alpha coefficient of 0.70 and higher is considered to be an acceptable threshold for reliability [27]. To obtain split-half reliability the function splithalf.r from the "multicon" R-package in the version 1.6 was used on the items of the postoperative QoR-PACU results [31].

Responsiveness refers to the ability of the instrument to detect change over time [32]. Responsiveness was analyzed considering pre- and postoperative QoR-PACU sum scores and was expressed with Cohen's effect size and standardized response mean. Cohen's effect size is defined as mean difference between preoperative and postoperative QoR-PACU sum scores divided by the SD of the preoperative QoR-PACU sum scores. Standardized response mean was calculated as the mean difference between pre- and postoperative QoR-PACU sum scores divided by the SD of these differences.

The percentage of successfully completed pre- and postoperative QoR-PACU questionnaires was used to assess feasibility. To evaluate acceptability of the QoR-PACU, we involved

patients and PACU staff and invited them to judge each item regarding importance and appropriateness.

All analyses were done on complete available cases, so no imputation of missing data was performed. P-values are presented as descriptive summary measures and do not represent results of confirmatory testing. No adjustment for multiplicity was performed. All analyses were performed with R Statistical Software, version 3.5.3 (R Foundation for Statistical Computing, Vienna, Austria).

## Results

### Demographic and clinical characteristics

Between March and November 2020, 255 patients were approached by the study team for the assessment with the QoR-PACU version 3. Nine patients were not available for the postoperative assessment in the PACU due to postponed surgeries. These patients were excluded from the analysis. A total of 246 patients completed the final version of the QoR-PACU questionnaire resulting in a completion rate of 96.5%. Fig 1 shows the flow of participants during the course of the study. Details on baseline demographic and clinical characteristics and perioperative variables related to surgery and anesthesia are presented in Table 1.

### QoR-PACU

Median sum scores of the four QoR-PACU assessments are presented in Table 2.

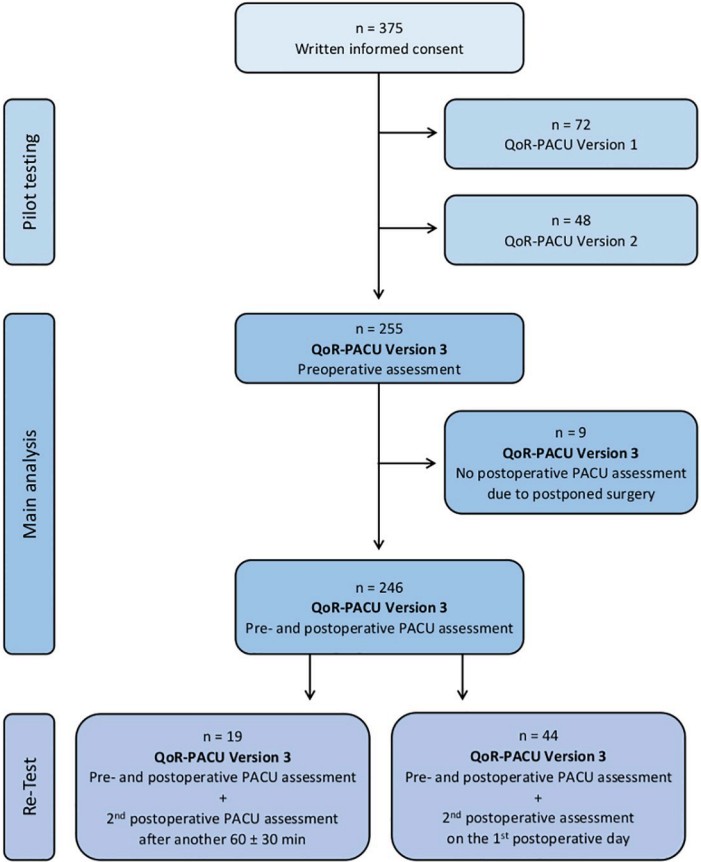

**Fig 1. The flow diagram shows patients included and excluded throughout the course of the study.**

**Table 1. Demographic and clinical characteristics.** Data are presented as median (25th; 75th percentile) or n (%). ASA: American Society of Anesthesiologists. PACU: postanaesthesia care unit.

| | n = 246 |
|---|---|
| **Patient characteristics** | |
| Age (years) | 64 (60; 69) |
| Body Mass Index (kg/m$^2$) | 26.5 (24.5; 28.9) |
| ASA physical status | |
| *II* | 214 (87) |
| *III* | 32 (13) |
| Charlson Comorbidity Index | 4 (4; 5) |
| Obstructive sleep apnea syndrome | 17 (7.6) |
| Education | |
| *<9 years* | 2 (0.8) |
| *9–10 years* | 28 (11.4) |
| *10–12 years* | 95 (38.6) |
| *12–13 years* | 16 (6.5) |
| *University degree* | 105 (42.7) |
| **Surgery** | |
| Duration of surgery (min) | 153.0 (135.0; 175.8) |
| Surgical approach | |
| *Open retropubic radical prostatectomy* | 104 (42.3) |
| *Robot-assisted radical prostatectomy* | 142 (57.7) |
| **Anesthesia and perioperative medication** | |
| Duration of anesthesia (min) | 223.5 (206.0; 247.8) |
| Premedication with Midazolam | 8 (3.3) |
| Antiemetic prophylaxis | |
| *None* | 2 (0.8) |
| *Dexamethasone* | 6 (2.4) |
| *Ondansetron* | 3 (1.2) |
| *Dexamethasone and Ondansetron* | 235 (95.5) |
| Anesthesia maintenance | |
| *Sevoflurane* | 241 (98.0) |
| *Propofol* | 5 (2.0) |
| Sufentanil (cumulative; μg) | 85.0 (70.0; 95.0) |
| Norepinehphrine (maximum dosage; μg/kg/min) | 0.1 (0.07; 0.14) |
| Fluids | |
| *Crystalloids (ml)* | 2500 (2000; 3000) |
| *Colloids (ml)* | 0 (0;0) |
| **Postoperative care and medication** | |
| Length of PACU stay (min) | 152.0 (118.3;196.5) |
| Piritramide (cumulative; mg) | 3.75 (3.75;7.5) |
| Pethidine (cumulative; mg) | 25.0 (25.0;25.0) |
| Discharge to | |
| *Normal ward* | 208 (84.6) |
| *Scheduled overnight PACU stay* | 27 (11.0) |
| *Unscheduled overnight PACU stay* | 11 (4.5) |

**Table 2. QoR-PACU scores for each assessment.** Data are presented as median (25th; 75th percentile). QoR: Quality of recovery, PACU: postanesthesia care unit.

| Assessment | n | Time | sum score |
|---|---|---|---|
| Preoperative | 246 | 24 ± 6 hrs before surgery | 50.0 (48.0; 51.0) |
| 1st PACU | 246 | 125.0 (83.0; 156.8) min after arrival in the PACU | 42.0 (38.0; 44.0) |
| 2nd PACU | 19 | 189.0 min (148.8; 215.8) min after arrival in the PACU | 45.0 (42.5; 47.0) |
| 24 hours postop | 44 | 24 ± 6 hrs after surgery | 45.0 (42.8; 47.3) |

Pre- and postoperative mean QoR-PACU scores for each item and the mean sum score are presented in Table 3. Fig 2 shows pre- and postoperative QoR-PACU scores.

## Validity, reliability, and responsiveness

Postoperative QoR-PACU sum scores did not differ across categories of clinically relevant variables: ASA physical status II (n = 214; 42 [38;45]) vs. ASA III (n = 32; 43 [40;44], p = 0.867). Low or intermediate risk for obstructive sleep apnea syndrome (n = 133; 41 [38;44]) vs. high risk or confirmed disease (n = 91; 42 [40;45], p = 0.134). The correlation between postoperative QoR-PACU sum scores and clinically relevant continuous variables is presented in Fig 3. Cronbach's alpha was 0.67 (95% CI: 0.61 to 0.73), reflecting moderate internal consistency [34]. The average of all split-half correlations was 0.52. The average of all split-half reliabilities was 0.69 ± 0.08. Interitem correlations and correlations between the QoR-PACU sum score and each item are presented in Fig 4 and S3 Table. There was a positive correlation between the QoR-PACU sum score and the score at the second assessment approximately one hour later (r = 0.71, 95%CI: 0.37 to 0.88, p < 0.01) reflecting acceptable test-retest reliability. Cohen's effect size and standardized response mean are presented in Table 3.

**Table 3. Mean QoR-PACU scores for each item and the mean QoR-PACU sum score.** Responsiveness is expressed with Cohen effect size (difference between preoperative and postoperative QoR-PACU scores, divided by the preoperative SD, 0.2 being considered small, 0.5 as medium, and 0.8 or greater as large [33]) and the standardized response mean (score difference divided by the SD of the score difference). Numbers are given as mean ± standard deviation (SD) unless otherwise indicated. QoR: Quality of recovery, PACU: postanesthesia care unit; CI: confidence interval.

| QoR-PACU item | Preoperative | Postoperative | Mean change [95% CI] | % change from baseline | Cohen effect size | Standardised response mean |
|---|---|---|---|---|---|---|
| 1. Pain | 3.7 ± 0.6 | 2.1 ± 1.2 | -1.6 [-1.8; -1.5] | 43.0 | 2.6 | 1.2 |
| 2. Sore throat | 3.8 ± 0.5 | 2.3 ± 1.2 | -1.6 [-1.7; -1.4] | 39.0 | 3.2 | 1.2 |
| 3. Dry mouth | 3.7 ± 0.6 | 1.7 ± 1.1 | -2.1 [-2.2; -1.9] | 54.0 | 3.4 | 1.7 |
| 4. Not able to breathe easy | 3.9 ± 0.4 | 3.6 ± 0.7 | -0.3 [-0.4; -0.2] | 7.7 | 0.7 | 0.4 |
| 5. Nausea | 4.0 ± 0.2 | 3.6 ± 0.8 | -0.3 [-0.4; -0.3] | 10.0 | 1.9 | 0.5 |
| 6. Vomiting | 4.0 ± 0.1 | 4.0 ± 0.2 | 0.0 [-0.1; 0.0] | 0.0 | 0.0 | 0.0 |
| 7. Feeling too cold | 3.9 ± 0.4 | 3.7 ± 0.7 | -0.2 [-0.3; -0.1] | 5.1 | 0.5 | 0.3 |
| 8. Feeling dizzy | 4.0 ± 0.2 | 2.4 ± 1.2 | -1.6 [-1.7; -1.4] | 40.0 | 9.6 | 1.4 |
| 9. Feeling confused | 4.0 ± 0.2 | 3.5 ± 0.9 | -0.5 [-0.6; -0.4] | 12.0 | 2.2 | 0.6 |
| 10. Feeling anxious | 3.4 ± 0.9 | 3.7 ± 0.6 | 0.3 [0.2; 0.5] | -8.8 | -0.4 | -0.3 |
| 11. Able to understand instructions or advice | 3.7 ± 0.5 | 3.7 ± 0.6 | -0.1 [-0.1; 0.0] | 0.0 | 0.0 | 0.0 |
| 12. Feeling comfortable | 3.4 ± 0.6 | 3.2 ± 0.8 | -0.2 [-0.3; -0.1] | 5.9 | 0.3 | 0.2 |
| 13. Getting support from hospital doctors and nurses | 3.8 ± 0.4 | 3.9 ± 0.4 | 0.0 [0.0; 0.1] | -2.6 | -0.2 | -0.2 |
| Sum | 49.0 ± 2.6 | 41.0 ± 5.0 | -8.0 [-8.6; -7.4] | 16.0 | 3.1 | 1.7 |

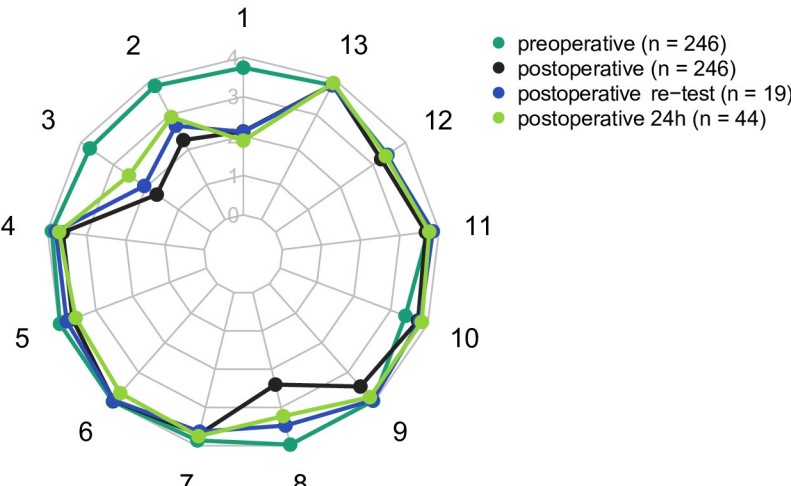

**Fig 2. The radar chart–spider diagram shows mean scores of single items of the QoR-PACU preoperatively (dark green, n = 246), in the PACU (black, n = 246), at re-assessment in the PACU (blue, n = 19), and on the day after surgery (light green, n = 44).** Each item of the questionnaire is presented as a spoke. The 5-point numeric rating scale is presented on the axis with numbers from 0 to 4.

## Discussion

The aim of this study was to establish a questionnaire to assess self-reported QoR during the recovery period after elective non-cardiac surgery. We developed the QoR-PACU based on the 40-item QoR-40 questionnaire. We tested the questionnaire in a cohort of 246 urological patients, 2 ± 1 hours after PACU admission. We found high acceptability and feasibility, good validity, adequate responsiveness, and moderate reliability.

A standardized tool for the assessment of patient-reported QoR in the PACU is urgently needed for both research and clinical purposes. Myles et al. emphasize that results of clinical research can only be considered valid if a reconfirmation is possible [8]. However, comparability and impact of clinical research is substantially diminished by different outcome definitions and the use of numerous instruments for psychometric assessment [8, 9, 35]. Therefore, it is important to standardize endpoints in clinical research.

For decades, the Aldrete scoring system has been widely used to determine, if a patient can be safely discharged from the PACU [20]. Items addressed by the Aldrete score are limited to

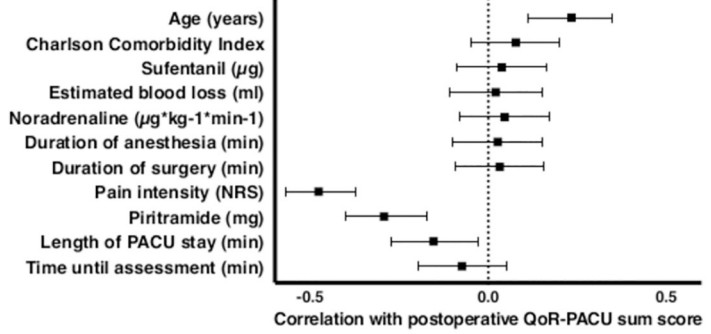

**Fig 3. Correlation between postoperative QoR-PACU sum scores and clinically relevant continuous variables.** Data are presented as Pearson's product-moment correlation coefficients and corresponding 95% confidence intervals.

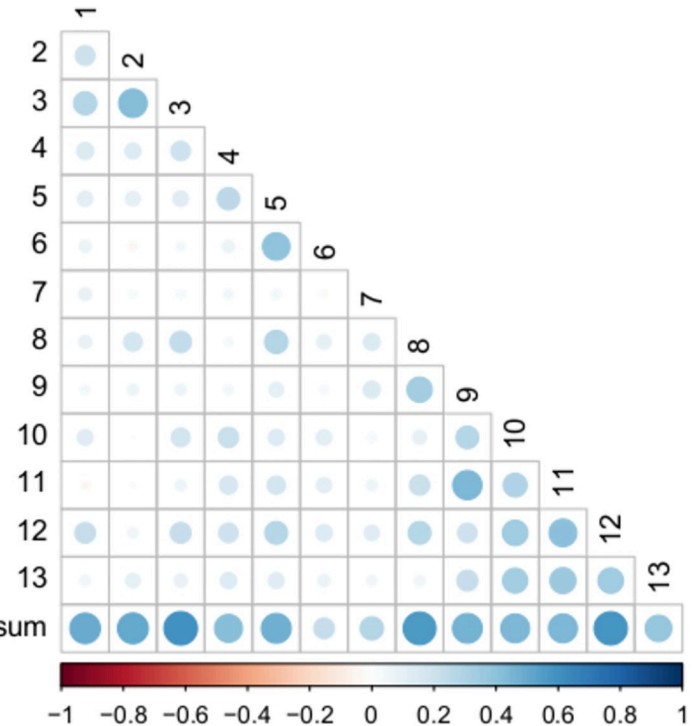

**Fig 4. Correlations between each item and the sum score and between single items of the postoperative QoR-PACU using Spearman correlation coefficients.**

physical aspects: activity, respiration, circulation, consciousness and color / oxygen saturation [21, 36]. However, the definition of adequate recovery may differ substantially between patients and caregivers. Including the patients' perception of postoperative recovery immediately after surgery provides a basis for the optimization of recovery, which may result in better outcomes and might even have a beneficial effect on length of stay and healthcare costs [3].

We chose clinically relevant aspects of intra- and postoperative care to assess construct validity. Pain intensity, piritramide dose, and PACU length of stay negatively correlated with QoR in the PACU. Our results confirm that pain perception plays a substantial role in perioperative care with a major impact on patients' perception of health status and recovery [37]. Despite homogeneity of surgical procedures, we found a large range of PACU length of stay. Shorter stay in the PACU correlated with better QoR, which confirms previous reports on the association between PACU length of stay and postoperative complications [38]. We assessed patients 120 ± 60 minutes after arrival in the PACU. This long period of recovery may have enhanced the understanding and completion of the questionnaire. However, in many anesthetic and surgical departments, it is common for patients to leave the PACU earlier [39]. Future studies should evaluate whether the QoR-PACU is also feasible earlier after arrival in the PACU.

Psychometric properties of the QoR-PACU revealed good validity and adequate responsiveness.

However, measures of reliability including internal consistency were moderate. The fact that the internal consistency of the QoR-PACU was not as high as expected is interesting, since all items of the QoR-PACU were derived from the QoR-40 which has been developed to evaluate the quality of recovery 24 hours after surgery and has been validated extensively.

Items showing high validity and reliability 24 hours after surgery showed only sufficient internal consistency in the immediate postoperative period. Several factors may account for the difference in internal consistency. First and most importantly, the patients' mental and physical condition changes rapidly during the early postoperative period. Of note, vigilance and pain perception are interconnected. When the effects of anesthesia and analgesia wear off, consciousness improves, and patients become more susceptible to postoperative pain. Therefore, it is reasonable to expect that self-perceived recovery in the early postoperative period will change substantially within a 30-minute time frame. Second, we noticed that the simultaneous application of measures of frequency, as used in the QoR-PACU, and measures of intensity, as used in the NRS, during the recovery period led to confusion with study participants. We tried to avoid this problem by reducing the 11-point scale to a 5-point scale linked with adverbs of frequency. Yet, difficulty in understanding measures of frequency might have influenced our results. Third, it is noteworthy that we observed a relatively small change from preoperative to PACU scores in our study population. The mean change from preoperative to PACU scores was <0.5 for 9 items, but not for the features pain, feeling confused, dry mouth, and sore throat. The majority of patients had a rather low perioperative risk as reflected by the ASA physical status. High perioperative risk has been found to be associated with poor recovery after colorectal cancer surgery [40]. Similarly, low ASA physical status might have contributed to the overall high QoR reported by participants of our study.

When evaluating responsiveness, it is commendable to compare the ability of the instrument to detect change to a gold standard [32, 41]. Unfortunately, in the present study this was not possible since there has not been any tool for the measurement of patient-reported recovery in the PACU to date. We followed the approach of Stark et al. and evaluated the ability of the QoR-PACU to detect change over time based on the Cohen's effect size and the standardized response mean [7]. Additionally, the correlation of QoR-PACU sum scores with clinically important parameters, such as patient-reported pain intensity confirms the clinical relevance of the observed change in QoR-PACU sum scores.

We found the application of the QoR-PACU during the recovery period feasible with high response and completion rates. Sum scores were highest at baseline on the day before surgery and lowest during assessment in the PACU followed by an increase on the first postoperative day. The development of QoR-PACU sum scores over time indicates that the QoR-PACU adequately mirrors QoR despite moderate internal consistency.

This validation study was performed at the PACU of a prostate cancer clinic. All surgical procedures and perioperative care at our prostate cancer center are highly standardized. Although allowing for excellent comparability between participants, generalizability is limited. We included solely male patients scheduled for radical prostatectomy. Results from previous studies suggest that gender aspects have an impact on postoperative QoR and speed of recovery [2, 42–44]. Overall, female patients tend to have lower QoR and longer PACU stay [2, 42, 44]. Moreover, pain intensity, nausea, and vomiting after surgery are more frequently reported by female patients [42, 43]. Gender aspects may be of high importance in individualized perioperative care and postoperative recovery.

We developed the QoR-PACU in German language to allow for evaluation of the instrument in our German speaking patient population. It is a natural limitation of any questionnaire that it can only be used with patients who have excellent understanding of the respective language. To be useful for a broader international public, the QoR-PACU needs to be translated into English and subsequently to various other languages. Although there are multiple instruments for assessing recovery 24 hours after the intervention, we have developed the QoR-PACU from the QoR-40 [2]. We have explicitly decided to derive the QoR-PACU from the QoR questionnaires, since their use has been recommended by the StEP initiative [9].

Future studies should evaluate the psychometric properties of the QoR-PACU in a more heterogenous patient population, including female and gender diverse patients, as well as a greater variety of patient-related and procedure-related risk factors. This might reveal whether the issue of moderate internal consistency was primarily linked to the characteristics of the initial study cohort, or whether items have to be revised substantially to be suitable for patients in the PACU. For the modification of the QoR-PACU, it might be helpful to consider suggestions from patients and caregivers.

## Conclusion

This preliminary validation study presents the development of a questionnaire to assess self-reported QoR after surgery in the PACU. We found high acceptability and feasibility, good validity, and adequate responsiveness. Against our hypothesis, we did not find high internal consistency. Based on these findings, the QoR-PACU should be modified. The modification process should also consider suggestions from healthcare professionals and patients. Future psychometric evaluation should include a more heterogeneous patient cohort including female and gender-diverse patients with varying degrees of perioperative risk.

## Supporting information

**S1 Table. QoR-PACU Version 3.0 preoperative.**
(DOCX)

**S2 Table. QoR-PACU Version 3.0 postoperative.**
(DOCX)

**S3 Table. Postoperative inter-item correlation.** Postoperative assessment: Inter-item correlations for the 13 items of the QoR-PACU score. Correlations are expressed as Pearson correlation coefficients.
(DOCX)

## Acknowledgments

The authors thank Lars Nawrath for his support in the process of selecting items of the QoR-40 questionnaire to develop the first version of the QoR-PACU.

All authors approved of the version to be published and agree to be accountable for all aspects of the work, thereby ensuring that questions related to the accuracy or integrity of any part of the work are appropriately investigated and resolved.

## Author Contributions

**Conceptualization:** Marlene Fischer.

**Data curation:** Ursula Kahl, Katrin Brodersen, Linda Krause, Marlene Fischer.

**Formal analysis:** Ursula Kahl, Linda Krause, Marlene Fischer.

**Investigation:** Ursula Kahl, Katrin Brodersen, Sarah Kaiser, Marlene Fischer.

**Project administration:** Katrin Brodersen, Marlene Fischer.

**Supervision:** Marlene Fischer.

**Visualization:** Linda Krause, Marlene Fischer.

**Writing – original draft:** Ursula Kahl, Katrin Brodersen, Marlene Fischer.

**Writing – review & editing:** Sarah Kaiser, Linda Krause, Regine Klinger, Lili Plümer, Christian Zöllner.

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
