## [Decision Letter · Decision Letter 0]

8 Aug 2022

PONE-D-22-18272Psychometric evaluation of a quality of recovery score for the postanesthesia care unit – a prospective validation studyPLOS ONE

Dear Dr. Marlene Fischer,

Thank you for submitting your manuscript to PLOS ONE. After careful consideration, we feel that it has merit but does not fully meet PLOS ONE’s publication criteria as it currently stands. Therefore, we invite you to submit a revised version of the manuscript that addresses the points raised during the review process.

Thank you for submitting interesting article. I believe our reviews will improve your paper.

We look forward to receiving your revised manuscript.

Kind regards,

Seung-Hwa Lee

Academic Editor

PLOS ONE

Journal Requirements:

Reviewers' comments:

Reviewer's Responses to Questions

**Comments to the Author**

1. Is the manuscript technically sound, and do the data support the conclusions?

Reviewer #1: Yes

Reviewer #2: Partly

2. Has the statistical analysis been performed appropriately and rigorously? 

Reviewer #1: Yes

Reviewer #2: Yes

3. Have the authors made all data underlying the findings in their manuscript fully available?

Reviewer #1: Yes

Reviewer #2: No

4. Is the manuscript presented in an intelligible fashion and written in standard English?

Reviewer #1: Yes

Reviewer #2: Yes

5. Review Comments to the Author

Reviewer #1: There is smaller sample size than calculated. It should be clarified. One more area is missing, the reasons that compelled the patients to fail to reach the follow up visits are to be evaulatied.

Otherwise, the article has merit to be accepted for publication.

Reviewer #2: Many thanks for sending this manuscript for the review. I have read it with great interest and you can find my suggestions for improvement below. In addition to those, my main concern is how a tool in German language could be of interest to wider international audience where English is the primary language. Another main concern is testing in PACU –120 minutes after PACU admission seems late as many PACU stays are much shorter for this surgical procedure.

Abstract:

Needs to be improved based on the comments below. Additionally add: refer to German tool, sample size is 255; others were in piloting stage of the tool development;

Introduction:

Overall, well described background. I suggest you improve the last sentence: "in patients after general anesthesia for elective non-cardiac surgery." – this is misleading as your tool was used only for patients undergoing urologic surgery and in German language only.

Material and methods:

- I wonder if quaternary (check spelling for this word please) hospital care is readily generalizable (see definitions of tertiary – quaternary care); your study was registered and there you mentioned tertiary care hospital (also the radical prostatectomy is a common procedure)

- This sentence is unclear, including chronologically – is this during the instrument development stage? How were the patients randomly selected? Or was this an opportunistic sample? Maybe this should be put in the next section? "We performed a pre-test of the QoR-PACU in a randomly selected cohort of 10 patients to assess feasibility."

- Please take care of writing this: 60 minutes ± 30 : should be 60 ± 30 minutes

- In the section: Development and adaptation of the QoR-PACU; I suggest you refer to pilot 1 and pilot 2 instead of early study period: "After a successful pretest phase, we administered the 16-item QoR-PACU (version 1) to 72 patients – the first pilot testing of the QoR-PACU." And similar where needed in this paragraph

- Sample Size: I suggest you use only 10 participants per item here and add that you aimed for at least 160 participants with full completion of the questionnaire version 3 (you did not reach 375 that you mention here). What I am confused about is several subdivision into subgroups for different testing... you should define sample sizes for all of these here or at least further develop figure 1 in results to show the numbers of patients doing reassessments

- I wonder if you chose the items from the QoR40 tool from English or valid German translation? Please clarify in the methods.

- Responsiveness definition: ability to detect clinical important change: what does this mean for your data? Add in table 2 which is important change?

- This is unclear: "The proportion of patients who successfully completed the QoR-PACU postoperatively was used to assess acceptability and feasibility." What proportion? How was this tested? Definitions?

Results:

- Please see above abour improving figure 1

- Table 2: i am ok with means here, however you should improve statistical part of the methods to reflect that as you say only medians with IQR will be reported; %change from baseline: be careful which should be negative – almost all? Also the paragraph before this table reports medians and then the table reports means – should be consistent; Also: the table title Responsiveness is not ok; please improve to XYZ and responsiveness.

- Figure 2: also add n of patients in brackets in the legend of the four colors; also improve figure 2 title

- Table 3: I suggest you remove this table completely as the construct validity in this context is not valid: your numbers of patients in each of the category are far from comparable in all clinically relevant variable as you say with the exception of N for ASA and OSAS (but I also disagree with the notion how your tool would be connected to mode of intubation for example). Also modify methods section appropriately

- Figure 3: I am fine with this one; please add what is presented in description: mean with 95% CI?? What statistical test was used for correlations – please also add this in the figure

Discussion:

- Some parts need to be modified to reflect my comments above including the first paragraph which needs to be significantly improved.

- How do you conclude this Is highly feasible and acceptable? Add the earliest 2 hours after PACU admission? Omit excellent recruitment this is ambiguous; leave high completion rates; also clarify in the methods section if patients alone completed the surveys or were the items read to them and explained?

- Your PACU times are interesting as in many countries patients mostly leave PACUs after around 60 minutes – please add to limitations how can your QoR instrument be used sooner;

- Use some citations when you refer to gender influences in the discussion

6. PLOS authors have the option to publish the peer review history of their article (what does this mean?). If published, this will include your full peer review and any attached files.

Reviewer #1: **Yes: **Shahjada Selim

Reviewer #2: No

---

## [Author Response · Author response to Decision Letter 0]

31 Oct 2022

Point by Point Response to the reviewers’ comments. 

We thank the reviewers for taking the time to read and process our manuscript. We have integrated the reviewers’ suggestions and believe that this has substantially improved the manuscript.

Please find below the reviewers’ comments written upright in black and our responses written in blue italics.

Reviewer #1:

1. There is smaller sample size than calculated. It should be clarified. 

We thank the reviewer for alerting us to this. According to the suggestions of both reviewer 1 and 2, we have amended the methods section for the following explanation: 

“There is no consistent recommendation regarding sample size for the development and the evaluation of a questionnaire. A “rule of thumb” suggests at least 10 participants for each scale item (Boateng et al. 2018; Nunnally 1978). This would result in 160 participants for the 16-items version 1, or 130 participants for the 13-items version 3 of the questionnaire. We did not a priori decide on the exact number of items to be included in the questionnaire. Instead, we planned to conduct a pilot phase to optimize the development of the questionnaire. Therefore, we opted for the inclusion of 375 patients, allowing for an extensive pilot testing sample as well as a large drop-out rate.

See revised article with highlighted changes, p.7

2. One more area is missing, the reasons that compelled the patients to fail to reach the follow up visits are to be evaulatied.

We thank the reviewer for alerting us to this. Due to postponed surgeries, 9 patients could not be met in the PACU by the study team. We have included this information in the flow chart (Fig 1).

See revised Fig 1

Reviewer #2: 

1. my main concern is how a tool in German language could be of interest to wider international audience where English is the primary language. 

We agree with the reviewer that in order to be useful to an international public, the QoR-PACU needs to be translated to English and subsequently to various more languages. It is a natural limitation that any questionnaire can only be used with patients who have excellent understanding of the respective language the questionnaire was written in. We developed the QoR-PACU in German language because we wanted to validate and use the instrument in our German speaking patient population. We have added these thoughts to the discussion section of the manuscript.

See revised article with highlighted changes, p.19

2. Another main concern is testing in PACU –120 minutes after PACU admission seems late as many PACU stays are much shorter for this surgical procedure.

We agree with the reviewer that this topic merits further discussion. 

In our study, the overall length of stay in our patient cohort was rather long with a median time of 152 minutes. Data on PACU length of stay varies widely across studies: 

• A retrospective analysis of 17.047 patients revealed a median PACU length of stay of 117 (IQR 82, 165) minutes (Weissman, Scemama, and Weiss 2019).

• A study investigating the relationship of pain and postoperative nausea and vomiting with the PACU length of stay in 9.603 patients, showed a mean PACU length of stay of 3.2 ± 2.3 hours (Ganter et al. 2014). 

• A study investigating whether BIS-guided anaesthesia reduced the incidence of awareness during surgery in adults, found a median PACU length of stay of 63 (40–95) min in 1.225 patients with BIS monitoring and 66 (40–100) min in 1.238 patients without BIS monitoring (Myles et al. 2004).

In order for the instrument to be suitable for patients with shorter PACU stays, it is necessary to re-evaluate the QoR-PACU earlier after arrival in the PACU.

We have included the following paragraph in the discussion:

“We assessed patients 120 ± 60 minutes after arrival in the PACU. This long period of recovery may have enhanced the understanding and completion of the questionnaire. However, in many anesthetic and surgical departments, it is common for patients to leave the PACU earlier (Myles et al. 2004). Future studies should evaluate whether the QoR-PACU is also feasible earlier after arrival in the PACU.”

See revised article with highlighted changes, p.17

Abstract:

Needs to be improved based on the comments below. Additionally add: 

- refer to German tool, 

We added this information to the abstract.

See revised article with highlighted changes, p.2

- sample size is 255, others were in piloting stage of the tool development;

We added the desired information: “We included 375 patients. After two piloting phases including 72 and 48 patients, respectively, we administered the final version of the QoR-PACU to 255 patients, with a completion rate of 96.5%.”

See revised article with highlighted changes, p.2

3. Introduction:

Overall, well described background. I suggest you improve the last sentence: "in patients after general anesthesia for elective non-cardiac surgery." – this is misleading as your tool was used only for patients undergoing urologic surgery and in German language only.

We have included the following sentence:” […] Therefore we developed the German QoR-PACU and evaluated its psychometric properties in a cohort of patients scheduled for elective urologic surgery.” 

See revised article with highlighted changes, p.4

4. Material and methods:

I wonder if quaternary (check spelling for this word please) hospital care is readily generalizable (see definitions of tertiary – quaternary care); your study was registered and there you mentioned tertiary care hospital (also the radical prostatectomy is a common procedure)

We thank the reviewer for alerting us to this. Our institution includes departments which provide quaternary care. However, we agree that the procedure of prostatectomy (although performed at a highly specialized prostate cancer center) is better referred to as tertiary care. We have changed this term accordingly.

See revised article with highlighted changes, p.5

 

5. This sentence is unclear, including chronologically

– is this during the instrument development stage? How were the patients randomly selected? Or was this an opportunistic sample? Maybe this should be put in the next section? "We performed a pre-test of the QoR-PACU in a randomly selected cohort of 10 patients to assess feasibility."

We agree with the reviewer that this sentence belongs to the development section. We have changed this in the manuscript. Additionally, we changed the term “randomly selected cohort” to “convenience sample”. 

See revised article with highlighted changes, p.6

6. Please take care of writing this: 60 minutes ± 30: should be 60 ± 30 minutes

We have changed this in the manuscript.

See revised article with highlighted changes, p.5

7. In the section: Development and adaptation of the QoR-PACU; I suggest you refer to pilot 1 and pilot 2 instead of early study period: "After a successful pretest phase, we administered the 16-item QoR-PACU (version 1) to 72 patients – the first pilot testing of the QoR-PACU." And similar where needed in this paragraph

We thank the reviewer for this suggestion. We have rephrased this in the manuscript. 

See revised article with highlighted changes, p.6

8. Sample Size: I suggest you use only 10 participants per item here and add that you aimed for at least 160 participants with full completion of the questionnaire version 3 (you did not reach 375 that you mention here). 

We are grateful for this pragmatic suggestion. We have changed the paragraph as follows to the following explanation: 

“There is no consistent recommendation regarding sample size for the development and the evaluation of a questionnaire. A “rule of thumb” suggests at least 10 participants for each scale item (Boateng et al. 2018; Nunnally 1978). This would result in 160 participants for the 16-items version 1, or 130 participants for the 13-items version 3 of the questionnaire. We did not a priori decide on the exact number of items to be included in the questionnaire. Instead, we planned to conduct a pilot phase to optimize the development of the questionnaire. Therefore, we opted for the inclusion of 375 patients, allowing for an extensive pilot testing sample as well as a large drop-out rate”

See revised article with highlighted changes, p.7

9. What I am confused about is several subdivision into subgroups for different testing... you should define sample sizes for all of these here or at least further develop figure 1 in results to show the numbers of patients doing reassessments

Following the suggestion of the reviewer, we now present these two subsets of patients in figure 1.

See Fig1 and table 2

10. I wonder if you chose the items from the QoR40 tool from English or valid German translation? Please clarify in the methods.

We have selected the 15 items from the validated German translation of the QoR-40 (Plöger 2008). We have now specified this in the Development section. 

See revised article with highlighted changes, p.6

11. Responsiveness definition: ability to detect clinical important change: what does this mean for your data? 

12. Add in table 2 which is important change?

We thank the reviewer for alerting us to this critical issue. Indeed, the term “clinically important change” is problematic, as this would require the comparison with a gold standard (Middel et al., n.d.; Terwee et al. 2007). Regarding the QoR-PACU this is not possible, since there hasn’t been any tool for the measurement of patient-reported recovery in the PACU to date. The correlation of QoR-PACU sum scores with clinically important parameters, such as patient reported pain intensity (r= -0.48, 95%CI: -0.57 to -0.37, p <0.001) confirms the clinical relevance of the observed change in QoR-PACU sum scores. Since we did not analyse the correlation of single items with clinically important parameters, we rephrased the responsiveness definition as follows: “Responsiveness refers to the ability of the instrument to detect change over time.” 

See revised article with highlighted changes, p.8

Additionally, we have added the following paragraph as a limitation:

“When evaluating responsiveness, it is commendable to compare the ability of the instrument to detect change to a gold standard (32,41). Unfortunately, in the present study this was not possible, since there hasn’t been any tool for the measurement of patient-reported recovery in the PACU to date. We followed the approach of Stark et al. and evaluated the ability of the QoR-PACU to detect change over time based on the Cohen’s effect size and the standardized response mean (7). Additionally, the correlation of QoR-PACU sum scores with clinically important parameters, such as patient-reported pain intensity confirms the clinical relevance of the observed change in QoR-PACU sum scores.”

See revised article with highlighted changes, p.18

To enable the appraisal of Cohen’s effect size, we’ve added the Cohen thresholds small (0.2), medium (0.5) and large (0.8) (Norman, Wyrwich, and Patrick 2007) to the legend of table 3. 

See revised table 3

13. This is unclear: "The proportion of patients who successfully completed the QoR-PACU postoperatively was used to assess acceptability and feasibility." What proportion? How was this tested? Definitions?

We have clarified this terminology throughout the manuscript.

We have assessed acceptability of the instrument based on interviews with patients and PACU staff during the piloting and the testing phase. We did not quantify results of these interviews; however, we implemented their feedback in the development of version 3 of the QoR PACU.

We used the percentage of successfully completed pre- and postoperative QoR-PACU questionnaires to assess feasibility. The completion rate was 96.5%. We therefore concluded, the assessment of patient reported recovery in the PACU with the QoR PACU is highly feasible

See revised article with highlighted changes, p.6, p.9

14. Results:

Please see above abour improving figure 1

We have revised figure 1.

See revised Fig 1

15. Table 2: 

- i am ok with means here, however you should improve statistical part of the methods to reflect that as you say only medians with IQR will be reported;

We have added this information in the statistical part of the methods.

See revised article with highlighted changes, p.8

- %Change from baseline: be careful which should be negative – almost all?

While the mean change is displayed as a negative or positive value, the % change from baseline is generally not displayed with an algebraic minus- or plus-sign. Please see for comparison Stark et al. (Stark, Myles, and Burke 2013).

 

- Also the paragraph before this table reports medians and then the table reports means – should be consistent; 

We decided to report mainly medians, because we believe that it is the most appropriate way to report questionnaire results. However, the calculation of the standardised response mean is based on mean values, hence the inconsistency between the text and the table. 

We have added the median sum scores of the four QoR-PACU assessments with the respective number of patients and time of assessment as Table 2.

See revised table 2

- Also: the table title Responsiveness is not ok; please improve to XYZ and responsiveness.

Following the approach of Stark et al. (Stark, Myles, and Burke 2013), we changed the title to “Change in mean QoR-PACU scores from preoperative to postoperative PACU assessment”.

See revised table 3

16. Figure 2: also add n of patients in brackets in the legend of the four colors; also improve figure 2 title

We included this information in the figure caption and changed the title to “Change in mean QoR-PACU scores”. 

See revised Fig 2

17. Table 3: I suggest you remove this table completely as the construct validity in this context is not valid: your numbers of patients in each of the category are far from comparable in all clinically relevant variable as you say with the exception of N for ASA and OSAS (but I also disagree with the notion how your tool would be connected to mode of intubation for example). Also modify methods section appropriately

We thank the reviewer for this suggestion. We have deleted this table. We now present information on the two variables ASA and OSAS in the main text, since we do not want to omit negative results. Additionally, this information might be interesting for comparison with future studies. 

See revised article with highlighted changes, p.13

18. Figure 3: I am fine with this one; please add what is presented in description: mean with 95% CI?? What statistical test was used for correlations – please also add this in the figure

We have included this information in the figure caption.

See revised Fig 3

19. Discussion:

Some parts need to be modified to reflect my comments above including the first paragraph which needs to be significantly improved.

- How do you conclude this Is highly feasible and acceptable? 

Please see above (Reviewer 2, Point 13)

See revised article with highlighted changes, p.6, p.9

- Add the earliest 2 hours after PACU admission? 

We added this information.

See revised article with highlighted changes, p.16

- Omit excellent recruitment this is ambiguous; leave high completion rates; 

We changed the sentence as follows: “We found high acceptability and feasibility, good validity, adequate responsiveness, and moderate reliability.”

See revised article with highlighted changes, p.16

- also clarify in the methods section if patients alone completed the surveys or were the items read to them and explained?

We clarified this section as follows: “Patients read and completed the questionnaire themselves. If necessary, patients were provided with glasses by the study team.”

See revised article with highlighted changes, p.5

- Your PACU times are interesting as in many countries patients mostly leave PACUs after around 60 minutes – please add to limitations how can your QoR instrument be used sooner;

Please see above (Reviewer 2, Point 2).

See revised article with highlighted changes, p.17

- Use some citations when you refer to gender influences in the discussion

We have added the appropriate citations: 

2. Myles PS, Weitkamp B, Jones K, Melick J, Hensen S. Validity and reliability of a postoperative quality of recovery score: the QoR-40. Br J Anaesth. 2000 Jan;84(1):11–5. 

39. Buchanan FF, Myles PS, Cicuttini F. Effect of patient sex on general anaesthesia and recovery. Br J Anaesth. 2011 Jun;106(6):832–9. 

40. Buchanan FF, Myles PS, Cicuttini F. Patient sex and its influence on general anaesthesia. Anaesth Intensive Care. 2009 Mar;37(2):207–18. 

41. Myles PS, McLeod AD, Hunt JO, Fletcher H. Sex differences in speed of emergence and quality of recovery after anaesthesia: cohort study. BMJ. 2001 Mar 24;322(7288):710–1.

See revised article with highlighted changes, p.18-19

 

References

Boateng, Godfred O., Torsten B. Neilands, Edward A. Frongillo, Hugo R. Melgar-Quiñonez, and Sera L. Young. 2018. ‘Best Practices for Developing and Validating Scales for Health, Social, and Behavioral Research: A Primer’. Frontiers in Public Health 6 (June): 149. https://doi.org/10.3389/fpubh.2018.00149.

Ganter, Michael T, Stephan Blumenthal, Seraina Dübendorfer, Simone Brunnschweiler, Tim Hofer, Richard Klaghofer, Andreas Zollinger, and Christoph K Hofer. 2014. ‘The Length of Stay in the Post-Anaesthesia Care Unit Correlates with Pain Intensity, Nausea and Vomiting on Arrival’. Perioperative Medicine 3 (November): 10. https://doi.org/10.1186/s13741-014-0010-8.

Middel, Berrie, Hanna Kuipers-Upmeijer, Jelte Bouma, Michiel Staal, Dettie Oenema, Theo Postma, Sijmon Terpstra, and Roy Stewart. n.d. ‘EVect of Intrathecal Baclofen Delivered by an Implanted Programmable Pump on Health Related Quality of Life in Patients with Severe Spasticity’, 6.

Myles, PS, K Leslie, J McNeil, A Forbes, and MTV Chan. 2004. ‘Bispectral Index Monitoring to Prevent Awareness during Anaesthesia: The B-Aware Randomised Controlled Trial’. The Lancet 363 (9423): 1757–63. https://doi.org/10.1016/S0140-6736(04)16300-9.

Norman, G. R., Kathleen W. Wyrwich, and Donald L. Patrick. 2007. ‘The Mathematical Relationship among Different Forms of Responsiveness Coefficients’. Quality of Life Research 16 (5): 815–22. https://doi.org/10.1007/s11136-007-9180-x.

Nunnally, Jum C. 1978. Psychometric Theory. 2d ed. McGraw-Hill Series in Psychology. New York: McGraw-Hill.

Plöger, Birgit. 2008. ‘Validierung einer deutschen Übersetzung des Quality-of-Recovery-Scores-40 (QoR-40) als Maß der patientenzentrierten postoperativen Ergebnisqualität’. Marburg: Philipps-Universität Marburg.

Sivan, Manoj. 2009. ‘Interpreting Effect Size to Estimate Responsiveness of Outcome Measures’. Stroke 40 (12): e709–e709. https://doi.org/10.1161/STROKEAHA.109.566836.

Stark, Peter A., Paul S. Myles, and Justin A. Burke. 2013. ‘Development and Psychometric Evaluation of a Postoperative Quality of Recovery Score’. Anesthesiology 118 (6): 1332–40. https://doi.org/10.1097/ALN.0b013e318289b84b.

Terwee, Caroline B., Sandra D. M. Bot, Michael R. de Boer, Daniëlle A. W. M. van der Windt, Dirk L. Knol, Joost Dekker, Lex M. Bouter, and Henrica C. W. de Vet. 2007. ‘Quality Criteria Were Proposed for Measurement Properties of Health Status Questionnaires’. Journal of Clinical Epidemiology 60 (1): 34–42. https://doi.org/10.1016/j.jclinepi.2006.03.012.

Weissman, Charles, Jeremy Scemama, and Yoram G. Weiss. 2019. ‘The Ratio of PACU Length-of-Stay to Surgical Duration: Practical Observations’. Acta Anaesthesiologica Scandinavica 63 (9): 1143–51. https://doi.org/10.1111/aas.13421.

---

## [Decision Letter · Decision Letter 1]

1 Jun 2023

PONE-D-22-18272R1Psychometric evaluation of a quality of recovery score for the postanesthesia care unit – a prospective validation studyPLOS ONE

Dear Dr. Marlene Fischer,

Thank you for submitting your manuscript to PLOS ONE. After careful consideration, we feel that it has merit but does not fully meet PLOS ONE’s publication criteria as it currently stands. Therefore, we invite you to submit a revised version of the manuscript that addresses the points raised during the review process.

I am especially sorry to late review. There's a lot of trouble to find reviewers.

We look forward to receiving your revised manuscript.

Kind regards,

Seunghwa Lee

Academic Editor

PLOS ONE

Reviewers' comments:

Reviewer's Responses to Questions

**Comments to the Author**

1. If the authors have adequately addressed your comments raised in a previous round of review and you feel that this manuscript is now acceptable for publication, you may indicate that here to bypass the “Comments to the Author” section, enter your conflict of interest statement in the “Confidential to Editor” section, and submit your "Accept" recommendation.

Reviewer #3: All comments have been addressed

Reviewer #4: All comments have been addressed

2. Is the manuscript technically sound, and do the data support the conclusions?

Reviewer #3: Yes

Reviewer #4: Yes

3. Has the statistical analysis been performed appropriately and rigorously? 

Reviewer #3: Yes

Reviewer #4: Yes

4. Have the authors made all data underlying the findings in their manuscript fully available?

Reviewer #3: Yes

Reviewer #4: Yes

5. Is the manuscript presented in an intelligible fashion and written in standard English?

Reviewer #3: Yes

Reviewer #4: Yes

6. Review Comments to the Author

Reviewer #3: First of all, I would like to thank you for the opportunity to review the article entitled "Psychometric evaluation of a quality of recovery score for the postanesthesia care unit – a prospective validation study" which I found of great interest.

I agree with most of the comments made by the previous reviewers and the authors have tried to address them.

As one of the reviewers commented, it is true that the main limitation is that the instrument is in German and adapted to the context of the country, but since there are no instruments for assessing health outcomes in perioperative patients, I consider its publication to be relevant, allowing later cultural adaptation to other languages.

As a general comment, they mention that there are multiple instruments for assessing recovery 24 hours after the intervention, but instead of reviewing these instruments or generating a pool of items, they decide to use the QoL-40, and it would be convenient to justify this aspect.

Regarding the abstract, perhaps it would be good to include the time in which the patients answered the instrument before and after the intervention (which seems to be about 2h after the intervention).

In the methods section, it would be good to indicate how the subsamples were selected to assess test-retest reliability, as well as for the pilot studies and why this sample size.

In the description of the sample size calculation, they indicate that they have included 375 participants, but in reality, a portion were for the pilot study and 255 participants were actually used to measure metric properties.

In the table 3 where they indicate the change in each item, it would be good to include the content of the item to assess the reason for the effect size.

They have also calculated Cronbach's alpha for the total instrument. An important part of the validation of an instrument is to assess its factor structure and to check the unidimensionality. I do not know if the authors initiated any analysis in this regard although they did not mention it in the article.

As a limitation, it should be pointed out that all the patients underwent radical prostatectomy surgery. This obviously implies that women have not been included, but also that the mean age of the sample will be mostly older adults.

Another limitation is that the patients were included in the development of the instrument but only for the pilot study or the final interviews. It is highly recommended to include patients and professionals in all phases of the development process, for example, through focus groups.

Although the article presents some preliminary analyses for the validation of the instrument, I believe that if the authors are able to address some comments and to emphasize the limitations by explaining that it is a preliminary validation study, the article is of interest.

Thank you

Reviewer #4: (No Response)

7. PLOS authors have the option to publish the peer review history of their article (what does this mean?). If published, this will include your full peer review and any attached files.

Reviewer #3: **Yes: **Yolanda Pardo Cladellas

Reviewer #4: No

---

## [Author Response · Author response to Decision Letter 1]

3 Jul 2023

PONE-D-22-18272R1 – Revision 2

Reviewer's Responses to Questions

Reviewer #3: 

1. As one of the reviewers commented, it is true that the main limitation is that the instrument is in German and adapted to the context of the country, but since there are no instruments for assessing health outcomes in perioperative patients, I consider its publication to be relevant, allowing later cultural adaptation to other languages.

With this study, we aimed to develop a new tool for the assessment of patient-reported quality of recovery (QoR) in the postanesthesia care unit (PACU). A questionnaire can only be used with patients who have excellent understanding of the respective language the questionnaire was written in. Therefore, to evaluate and to use the instrument in our German speaking patient population, we developed the QoR-PACU in German language.

We hope that international psychometricians and clinicians will find the QoR-PACU useful and will translate it into other languages. 

We have added these considerations to the discussion section of the manuscript.

See revised article with highlighted changes, p.19

2. As a general comment, they mention that there are multiple instruments for assessing recovery 24 hours after the intervention, but instead of reviewing these instruments or generating a pool of items, they decide to use the QoL-40, and it would be convenient to justify this aspect. 

We agree with the reviewer that using a larger pool of items to choose from, might have benefited the development of the QoR-PACU. Therefore, we have added this aspect as a limitation. Yet, we have explicitly decided to derive the QoR-PACU from the QoR questionnaires, since their use has been recommended by the StEP inititiative [1]. Future studies should include suggestions by anaesthesiologists, PACU nurses and patients.

Revised manuscript with highlighted changes, p.19

3. Regarding the abstract, perhaps it would be good to include the time in which the patients answered the instrument before and after the intervention (which seems to be about 2h after the intervention).

We have added this information to the abstract.

Revised manuscript with highlighted changes, p.2

4. In the methods section, it would be good to indicate how the subsamples were selected to assess test-retest reliability, as well as for the pilot studies and why this sample size.

Pilot study selection

The patients who were included in the pilot testing phase were selected consecutively in analogy to the patients included in the main analysis. All patients who met the inclusion criteria and did not meet exclusion criteria were approached by a member of the study team for participation in the study. 

Pilot study sample size

We did not calculate sample sizes for the pilot study. Instead, the pilot studies were conducted in exactly the same way as the main study. The pilot phase was halted when we encountered problems that led us to modify the questionnaire. 

The first pilot phase was halted after 72 patients, when we noticed misunderstandings regarding similarity between the 11-point response scale (measure of frequency) and the numeric rating scale used for assessment of pain (measure of intensity). The second pilot phase was halted after 48 patients, when we noticed that the font size of the questionnaire was too small for some patients.

Test-retest subgroup selection

To assess test-retest reliability, 19 patients completed the QoR-PACU twice, 60 ± 30 minutes after the first assessment. The two assessments were conducted by two different raters. On 5 consecutive days, two members of the study team were available for testing. During these days, all study participants underwent a second assessment and were included for test-retest reliability.

Test-Retest sample size

We aimed to analyze test-retest reliability in 20-25 patients [2–4]. As explained above, we performed retest on 5 consecutive days, enrolling a convenience sample of patients available on the respective days. 

5. In the description of the sample size calculation, they indicate that they have included 375 participants, but in reality, a portion were for the pilot study and 255 participants were actually used to measure metric properties.

We thank the reviewer for alerting us to this misleading wording and we have rephrased the paragraph as follows:

“There is no consistent recommendation regarding sample size for the development and the evaluation of a questionnaire. A “rule of thumb” suggests at least 10 participants for each scale item [27,28]. This would result in 160 participants for the 16-item version 1, or 130 participants for the 13-item version 3 of the questionnaire. We did not a priori decide on the exact number of items to be included in the questionnaire. Instead, we planned to conduct a pilot phase with at least 100 patients to optimize the development of the questionnaire. For the final analysis, we aimed to include at least 200 participants. Since we expected a drop-out rate of 20%, we opted for the overall inclusion of 375 patients.”

Revised manuscript with highlighted changes, p.7

6. In the table 3 where they indicate the change in each item, it would be good to include the content of the item to assess the reason for the effect size.

Thank you for this advice. We have added the item description to Table 3, p. 13. 

7. They have also calculated Cronbach's alpha for the total instrument. An important part of the validation of an instrument is to assess its factor structure and to check the unidimensionality. I do not know if the authors initiated any analysis in this regard although they did not mention it in the article.

Indeed, we have performed confirmatory and exploratory factor analyses. The exploratory factor analysis showed the presence of at least four factors (based on the p-value of the chi-squared test which was smaller than 5% for three factors and 0.112 for four factors). Those factors were not congruent with the questionnaire from which we adapted the items which had two dimensions, a physical and a mental dimension. Since our questionnaire is based on a validated questionnaire with two dimensions, we additionally performed confirmatory factor analysis, checking whether our adapted questionnaire shares the same dimensions. The confirmatory factor analysis showed a root mean squared error of approximation of 0.085 (90% CI: 0.07 – 0.10), which does not show a good fit. In the same line, the other indices show a rather unsatisfactory fit. Also, removing some of the items with a small loading did not improve the result. Since the results of the factor analyses do not add to the psychometric robustness of the QoR-PACU, but rather point to the necessity of further modifications of the instrument, we did not report these results but instead aim to continue improving the questionnaire. 

If requested by the editor or the reviewer, we will be happy to include the results of the factor analyses in the manuscript or as supplementary information. 

8. As a limitation, it should be pointed out that all the patients underwent radical prostatectomy surgery. This obviously implies that women have not been included, but also that the mean age of the sample will be mostly older adults.

We strongly agree with the reviewer. The fact that we included solely male patients of similar age and ethnicity, scheduled for the same surgical procedure, is one of the main limitations of our study.

We have mentioned this limitation in the manuscript:

“This validation study was performed at the PACU of a prostate cancer clinic. All surgical procedures and perioperative care at our prostate cancer center are highly standardized. Although allowing for excellent comparability between participants, generalizability is limited. We included solely male patients scheduled for radical prostatectomy. Results from previous studies suggest that gender aspects have an impact on postoperative QoR and speed of recovery [2,42–44]. Overall, female patients tend to have lower QoR and longer PACU stay [2,42,44]. Morevover, pain intensity, nausea, and vomiting after surgery are more frequently reported by female patients [42,43]. Gender aspects may be of high importance in individualized perioperative care and postoperative recovery.

[…]

Future studies should evaluate the psychometric properties of the QoR-PACU in a more heterogenous patient population, including female and gender diverse patients, as well as a greater variety of patient-related and procedure-related risk factors.”

We aim to conduct a follow-up study which will include a more heterogenous study cohort including female as well as gender-diverse patients, different age groups and a variety of surgical procedures.

See revised article with highlighted changes, p.18-19

9. Another limitation is that the patients were included in the development of the instrument but only for the pilot study or the final interviews. It is highly recommended to include patients and professionals in all phases of the development process, for example, through focus groups.

We agree with the reviewer, that it would be commendable to include patients in all phases of development, including the very early stages. We did not develop the QoR-PACU from scratch. Instead, we selected items from the QoR-40. At this stage of development, only professionals (experienced anesthesiologists) were involved. 

For the follow-up study we will modify the QoR-PACU. The modification will comprise suggestions from anesthesiologists, PACU nurses and patients.

We have modified the conclusion as follows:

“This preliminary validation study presents the development of a questionnaire to assess self-reported QoR after surgery in the PACU. We found high acceptability and feasibility, good validity, and adequate responsiveness. Against our hypothesis, we did not find high internal consistency. Based on these findings, the QoR-PACU should be modified. The modification process should consider suggestions from healthcare professionals and patients. Future psychometric evaluation should include a more heterogeneous patient cohort including female and gender-diverse patients with varying degrees of perioperative risk.”

See revised article with highlighted changes, p.19-20

10. Although the article presents some preliminary analyses for the validation of the instrument, I believe that if the authors are able to address some comments and to emphasize the limitations by explaining that it is a preliminary validation study, the article is of interest.

We thank the reviewer for this appraisal of our study. We have specified the term “preliminary validation study” in the conclusion (see comment above). To address the nature of the study more precisely as expressed by the reviewer, we suggest to edit the title of our manuscript into “Psychometric evaluation of a quality of recovery score for the postanesthesia care unit – a preliminary validation study

See revised article with highlighted changes, p.19-20

References

1. Myles PS, Grocott MPW, Boney O, Moonesinghe SR, Group C-S. Standardizing end points in perioperative trials: towards a core and extended outcome set. Br J Anaesth. 2016;116: 586–589. doi:10.1093/bja/aew066

2. Demumieux F, Ludes P-O, Diemunsch P, Bennett-Guerrero E, Lujic M, Lefebvre F, et al. Validation of the translated Quality of Recovery-15 questionnaire in a French-speaking population. Br J Anaesth. 2020;124: 761–767. doi:10.1016/j.bja.2020.03.011

3. Lyckner S, Böregård I-L, Zetterlund E-L, Chew MS. Validation of the Swedish version of Quality of Recovery score -15: a multicentre, cohort study. Acta Anaesthesiol Scand. 2018;62: 893–902. doi:10.1111/aas.13086

4. Stark PA, Myles PS, Burke JA. Development and Psychometric Evaluation of a Postoperative Quality of Recovery Score. Anesthesiology. 2013;118: 1332–1340. doi:10.1097/aln.0b013e318289b84b

---

## [Editor Report · Decision Letter 2]

19 Jul 2023

PONE-D-22-18272R2Psychometric evaluation of a quality of recovery score for the postanesthesia care unit – a preliminary validation studyPLOS ONE

Dear Dr. Fischer,

Thank you for submitting your manuscript to PLOS ONE. After careful consideration, we feel that it has merit but does not fully meet PLOS ONE’s publication criteria as it currently stands. Therefore, we invite you to submit a revised version of the manuscript that addresses the points raised during the review process.

We look forward to receiving your revised manuscript.

Kind regards,

Seunghwa Lee

Academic Editor

PLOS ONE

Journal Requirements:

Additional Editor Comments:

First of all, I would like to thank you for the opportunity to review the article entitled "Psychometric evaluation of a quality of recovery score for the postanesthesia care unit – a prospective validation study" which I found of great interest.

I agree with most of the comments made by the previous reviewers and the authors have tried to address them.

As one of the reviewers commented, it is true that the main limitation is that the instrument is in German and adapted to the context of the country, but since there are no instruments for assessing health outcomes in perioperative patients, I consider its publication to be relevant, allowing later cultural adaptation to other languages.

As a general comment, they mention that there are multiple instruments for assessing recovery 24 hours after the intervention, but instead of reviewing these instruments or generating a pool of items, they decide to use the QoL-40, and it would be convenient to justify this aspect.

Regarding the abstract, perhaps it would be good to include the time in which the patients answered the instrument before and after the intervention (which seems to be about 2h after the intervention).

In the methods section, it would be good to indicate how the subsamples were selected to assess test-retest reliability, as well as for the pilot studies and why this sample size.

In the description of the sample size calculation, they indicate that they have included 375 participants, but in reality, a portion were for the pilot study and 255 participants were actually used to measure metric properties.

In the table 3 where they indicate the change in each item, it would be good to include the content of the item to assess the reason for the effect size.

They have also calculated Cronbach's alpha for the total instrument. An important part of the validation of an instrument is to assess its factor structure and to check the unidimensionality. I do not know if the authors initiated any analysis in this regard although they did not mention it in the article.

As a limitation, it should be pointed out that all the patients underwent radical prostatectomy surgery. This obviously implies that women have not been included, but also that the mean age of the sample will be mostly older adults.

Another limitation is that the patients were included in the development of the instrument but only for the pilot study or the final interviews. It is highly recommended to include patients and professionals in all phases of the development process, for example, through focus groups.

Although the article presents some preliminary analyses for the validation of the instrument, I believe that if the authors are able to address some comments and to emphasize the limitations by explaining that it is a preliminary validation study, the article is of interest.

Thank you
---

## [Author Response · Author response to Decision Letter 2]

21 Jul 2023

Reviewer's Responses to Questions

Reviewer #3: 

1. As one of the reviewers commented, it is true that the main limitation is that the instrument is in German and adapted to the context of the country, but since there are no instruments for assessing health outcomes in perioperative patients, I consider its publication to be relevant, allowing later cultural adaptation to other languages.

With this study, we aimed to develop a new tool for the assessment of patient-reported quality of recovery (QoR) in the postanesthesia care unit (PACU). A questionnaire can only be used with patients who have excellent understanding of the respective language the questionnaire was written in. Therefore, to evaluate and to use the instrument in our German speaking patient population, we developed the QoR-PACU in German language.

We hope that international psychometricians and clinicians will find the QoR-PACU useful and will translate it into other languages. 

We have added these considerations to the discussion section of the manuscript.

See revised article with highlighted changes, p.19

2. As a general comment, they mention that there are multiple instruments for assessing recovery 24 hours after the intervention, but instead of reviewing these instruments or generating a pool of items, they decide to use the QoL-40, and it would be convenient to justify this aspect. 

We agree with the reviewer that using a larger pool of items to choose from, might have benefited the development of the QoR-PACU. Therefore, we have added this aspect as a limitation. Yet, we have explicitly decided to derive the QoR-PACU from the QoR questionnaires, since their use has been recommended by the StEP inititiative [1]. Future studies should include suggestions by anaesthesiologists, PACU nurses and patients.

Revised manuscript with highlighted changes, p.19

3. Regarding the abstract, perhaps it would be good to include the time in which the patients answered the instrument before and after the intervention (which seems to be about 2h after the intervention).

We have added this information to the abstract.

Revised manuscript with highlighted changes, p.2

4. In the methods section, it would be good to indicate how the subsamples were selected to assess test-retest reliability, as well as for the pilot studies and why this sample size.

Pilot study selection

The patients who were included in the pilot testing phase were selected consecutively in analogy to the patients included in the main analysis. All patients who met the inclusion criteria and did not meet exclusion criteria were approached by a member of the study team for participation in the study. 

Pilot study sample size

We did not calculate sample sizes for the pilot study. Instead, the pilot studies were conducted in exactly the same way as the main study. The pilot phase was halted when we encountered problems that led us to modify the questionnaire. 

The first pilot phase was halted after 72 patients, when we noticed misunderstandings regarding similarity between the 11-point response scale (measure of frequency) and the numeric rating scale used for assessment of pain (measure of intensity). The second pilot phase was halted after 48 patients, when we noticed that the font size of the questionnaire was too small for some patients.

Test-retest subgroup selection

To assess test-retest reliability, 19 patients completed the QoR-PACU twice, 60 ± 30 minutes after the first assessment. The two assessments were conducted by two different raters. On 5 consecutive days, two members of the study team were available for testing. During these days, all study participants underwent a second assessment and were included for test-retest reliability.

Test-Retest sample size

We aimed to analyze test-retest reliability in 20-25 patients [2–4]. As explained above, we performed retest on 5 consecutive days, enrolling a convenience sample of patients available on the respective days. 

5. In the description of the sample size calculation, they indicate that they have included 375 participants, but in reality, a portion were for the pilot study and 255 participants were actually used to measure metric properties.

We thank the reviewer for alerting us to this misleading wording and we have rephrased the paragraph as follows:

“There is no consistent recommendation regarding sample size for the development and the evaluation of a questionnaire. A “rule of thumb” suggests at least 10 participants for each scale item [27,28]. This would result in 160 participants for the 16-item version 1, or 130 participants for the 13-item version 3 of the questionnaire. We did not a priori decide on the exact number of items to be included in the questionnaire. Instead, we planned to conduct a pilot phase with at least 100 patients to optimize the development of the questionnaire. For the final analysis, we aimed to include at least 200 participants. Since we expected a drop-out rate of 20%, we opted for the overall inclusion of 375 patients.”

Revised manuscript with highlighted changes, p.7

6. In the table 3 where they indicate the change in each item, it would be good to include the content of the item to assess the reason for the effect size.

Thank you for this advice. We have added the item description to Table 3, p. 13. 

7. They have also calculated Cronbach's alpha for the total instrument. An important part of the validation of an instrument is to assess its factor structure and to check the unidimensionality. I do not know if the authors initiated any analysis in this regard although they did not mention it in the article.

Indeed, we have performed confirmatory and exploratory factor analyses. The exploratory factor analysis showed the presence of at least four factors (based on the p-value of the chi-squared test which was smaller than 5% for three factors and 0.112 for four factors). Those factors were not congruent with the questionnaire from which we adapted the items which had two dimensions, a physical and a mental dimension. Since our questionnaire is based on a validated questionnaire with two dimensions, we additionally performed confirmatory factor analysis, checking whether our adapted questionnaire shares the same dimensions. The confirmatory factor analysis showed a root mean squared error of approximation of 0.085 (90% CI: 0.07 – 0.10), which does not show a good fit. In the same line, the other indices show a rather unsatisfactory fit. Also, removing some of the items with a small loading did not improve the result. Since the results of the factor analyses do not add to the psychometric robustness of the QoR-PACU, but rather point to the necessity of further modifications of the instrument, we did not report these results but instead aim to continue improving the questionnaire. 

If requested by the editor or the reviewer, we will be happy to include the results of the factor analyses in the manuscript or as supplementary information. 

8. As a limitation, it should be pointed out that all the patients underwent radical prostatectomy surgery. This obviously implies that women have not been included, but also that the mean age of the sample will be mostly older adults.

We strongly agree with the reviewer. The fact that we included solely male patients of similar age and ethnicity, scheduled for the same surgical procedure, is one of the main limitations of our study.

We have mentioned this limitation in the manuscript:

“This validation study was performed at the PACU of a prostate cancer clinic. All surgical procedures and perioperative care at our prostate cancer center are highly standardized. Although allowing for excellent comparability between participants, generalizability is limited. We included solely male patients scheduled for radical prostatectomy. Results from previous studies suggest that gender aspects have an impact on postoperative QoR and speed of recovery [2,42–44]. Overall, female patients tend to have lower QoR and longer PACU stay [2,42,44]. Morevover, pain intensity, nausea, and vomiting after surgery are more frequently reported by female patients [42,43]. Gender aspects may be of high importance in individualized perioperative care and postoperative recovery.

[…]

Future studies should evaluate the psychometric properties of the QoR-PACU in a more heterogenous patient population, including female and gender diverse patients, as well as a greater variety of patient-related and procedure-related risk factors.”

We aim to conduct a follow-up study which will include a more heterogenous study cohort including female as well as gender-diverse patients, different age groups and a variety of surgical procedures.

See revised article with highlighted changes, p.18-19

9. Another limitation is that the patients were included in the development of the instrument but only for the pilot study or the final interviews. It is highly recommended to include patients and professionals in all phases of the development process, for example, through focus groups.

We agree with the reviewer, that it would be commendable to include patients in all phases of development, including the very early stages. We did not develop the QoR-PACU from scratch. Instead, we selected items from the QoR-40. At this stage of development, only professionals (experienced anesthesiologists) were involved. 

For the follow-up study we will modify the QoR-PACU. The modification will comprise suggestions from anesthesiologists, PACU nurses and patients.

We have modified the conclusion as follows:

“This preliminary validation study presents the development of a questionnaire to assess self-reported QoR after surgery in the PACU. We found high acceptability and feasibility, good validity, and adequate responsiveness. Against our hypothesis, we did not find high internal consistency. Based on these findings, the QoR-PACU should be modified. The modification process should consider suggestions from healthcare professionals and patients. Future psychometric evaluation should include a more heterogeneous patient cohort including female and gender-diverse patients with varying degrees of perioperative risk.”

See revised article with highlighted changes, p.19-20

10. Although the article presents some preliminary analyses for the validation of the instrument, I believe that if the authors are able to address some comments and to emphasize the limitations by explaining that it is a preliminary validation study, the article is of interest.

We thank the reviewer for this appraisal of our study. We have specified the term “preliminary validation study” in the conclusion (see comment above). To address the nature of the study more precisely as expressed by the reviewer, we suggest to edit the title of our manuscript into “Psychometric evaluation of a quality of recovery score for the postanesthesia care unit – a preliminary validation study

See revised article with highlighted changes, p.19-20

References

1. Myles PS, Grocott MPW, Boney O, Moonesinghe SR, Group C-S. Standardizing end points in perioperative trials: towards a core and extended outcome set. Br J Anaesth. 2016;116: 586–589. doi:10.1093/bja/aew066

2. Demumieux F, Ludes P-O, Diemunsch P, Bennett-Guerrero E, Lujic M, Lefebvre F, et al. Validation of the translated Quality of Recovery-15 questionnaire in a French-speaking population. Br J Anaesth. 2020;124: 761–767. doi:10.1016/j.bja.2020.03.011

3. Lyckner S, Böregård I-L, Zetterlund E-L, Chew MS. Validation of the Swedish version of Quality of Recovery score -15: a multicentre, cohort study. Acta Anaesthesiol Scand. 2018;62: 893–902. doi:10.1111/aas.13086

4. Stark PA, Myles PS, Burke JA. Development and Psychometric Evaluation of a Postoperative Quality of Recovery Score. Anesthesiology. 2013;118: 1332–1340. doi:10.1097/aln.0b013e318289b84b

---

## [Decision Letter · Decision Letter 3]

25 Jul 2023

Psychometric evaluation of a quality of recovery score for the postanesthesia care unit – a preliminary validation study

PONE-D-22-18272R3

Dear Dr. Fischer,

We’re pleased to inform you that your manuscript has been judged scientifically suitable for publication and will be formally accepted for publication once it meets all outstanding technical requirements.

Kind regards,

Seunghwa Lee

Academic Editor

PLOS ONE

Additional Editor Comments (optional):

Reviewers' comments:

Reviewer's Responses to Questions

**Comments to the Author**

1. If the authors have adequately addressed your comments raised in a previous round of review and you feel that this manuscript is now acceptable for publication, you may indicate that here to bypass the “Comments to the Author” section, enter your conflict of interest statement in the “Confidential to Editor” section, and submit your "Accept" recommendation.

Reviewer #3: All comments have been addressed

2. Is the manuscript technically sound, and do the data support the conclusions?

Reviewer #3: Yes

3. Has the statistical analysis been performed appropriately and rigorously? 

Reviewer #3: Yes

4. Have the authors made all data underlying the findings in their manuscript fully available?

Reviewer #3: Yes

5. Is the manuscript presented in an intelligible fashion and written in standard English?

Reviewer #3: Yes

6. Review Comments to the Author

Reviewer #3: I believe that the authors have addressed most of the comments I made. I would only suggest that the factor analysis be part of the results, even if only as additional material, as I believe it may help future researchers interested in the validation of the instrument to assess the structure of the instrument. Thank you for the opportunity to carry out this review.

7. PLOS authors have the option to publish the peer review history of their article (what does this mean?). If published, this will include your full peer review and any attached files.

Reviewer #3: No

---

## [Editor Report · Acceptance letter]

2 Aug 2023

PONE-D-22-18272R3 

Psychometric evaluation of a quality of recovery score for the postanesthesia care unit – a preliminary validation study 

Dear Dr. Fischer:

I'm pleased to inform you that your manuscript has been deemed suitable for publication in PLOS ONE. Congratulations! Your manuscript is now with our production department. 

Kind regards, 

on behalf of

Dr. Seunghwa Lee 

Academic Editor

PLOS ONE